# Effects of Below-Ground Microbial Biostimulant *Trichoderma harzianum* on Diseases, Insect Community, and Plant Performance in *Cucurbita pepo* L. under Open Field Conditions

**DOI:** 10.3390/microorganisms10112242

**Published:** 2022-11-12

**Authors:** Pierluigi Forlano, Stefania Mirela Mang, Vittoria Caccavo, Paolo Fanti, Ippolito Camele, Donatella Battaglia, Vincenzo Trotta

**Affiliations:** 1School of Agricultural, Forestry, Food and Environmental Sciences (SAFE), University of Basilicata, Viale dell’Ateneo Lucano 10, 85100 Potenza, Italy; 2Department of Science, University of Basilicata, Viale dell’Ateneo Lucano 10, 85100 Potenza, Italy

**Keywords:** zucchini squash, plant diseases, viruses, powdery mildew, aphids, parasitoids, integrated pest management

## Abstract

Agrochemicals are generally used in agriculture to maximize yields and product quality, but their overuse can cause environmental pollution and human health problems. To reduce the off-farm input of chemicals, numerous biostimulant products based on beneficial symbiont plant fungi are receiving a great deal of attention. The evolution of plant diseases and the performance of insects are influenced by plant chemical defences, both of which are, in turn, influenced by below-ground symbionts. Direct and indirect plant defences mediated by belowground symbionts against plant diseases and insect herbivores were demonstrated in greenhouses experiments. However, little attention has been paid to the use of *Trichoderma* under open field conditions, and no data are available for zucchini (*Cucurbita pepo* L.) plants in the field. To determine the effects of a commercial *Trichoderma harzianum* strain T22 on plant viruses, powdery mildew, the arthropod community, and on the agronomic performance associated with zucchini plants, an experiment was conducted in 2022 under open field conditions in South Italy. Our results indicate that *T. harzianum* T22 makes zucchini plants more attractive to aphids and to Hymenoptera parasitoid but failed to control zucchini pathogens. The complex plant–disease–arthropod–microorganism interactions that occurred in the field during the entire plant cycle are discussed to enrich our current information on the possibilities of using these microorganisms as a green alternative in agriculture.

## 1. Introduction

Agrochemicals are at the base of intensive agricultural systems, and the increasing demand of food for humans has enhanced their use worldwide [1]. Agrochemicals include pesticides and other product categories that promote plant growth and preserve plant health are used to maximize crop yield. However, the use of agrochemicals, particularly synthetic agrochemicals and inorganic fertilizers, causes toxicity to humans and ecosystems [2,3]. Another problem related to the use of synthetic agrochemicals is the increasing emergence of resistant strains of pests and pathogens [4,5,6].

Beneficial soil microbes, enhancing crop yield and promoting plant defences, such as non-pathogenic bacteria [7], mycorrhizal fungi [8] and plant-growth-promoting fungi [9], are a possible alternative to the use of agrochemicals. The application of beneficial microbial inoculants in agriculture has increased over the past two decades [10]. Fungi of the genus *Trichoderma* are among the most effective plant growth promoters in cultivated plant species [11]. The induction of plant resistance against pests and pathogens by fungi of the genus *Trichoderma* has been much studied in the tomato [12,13,14,15,16,17,18,19]. Some *Trichoderma* strains were found to activate the plant systemic acquired resistance (SAR) and/or induce systemic resistance (ISR) against biotic and abiotic stress agents [11,20,21,22,23]. For example, tomato defence responses against the green stink bug *Nezara viridula* L. were enhanced by *T. harzianum* strain T22 through an early increase in transcript levels of jasmonic acid (JA) marker genes [14].

Zucchini (*Cucurbita pepo* L.) is the most important economically and globally widespread species among the cultivated Cucurbitaceae [24]. Zucchini includes a wide assortment of varieties and cultivars [25] and is one of the most important and consumed vegetables worldwide. However, there are few research studies investigating the interaction between *Trichoderma* spp. and zucchini pathogens [26,27].

In the field, aphids, mainly *Aphis gossypii* Glover (Homoptera: Aphididae), and some pathogens, such as phytoviruses and the powdery mildew fungal agents, namely, *Golovinomyces cichoracearum* (DC.) V.P. Heluta, *Podosphaera fusca* (Fr.) U. Braun & Shishkoff and *Leveillula taurica* (Lév.) G. Arnaud, are the most harmful organisms that cause plant damage and production losses in zucchini crops [28,29,30]. *Aphis gossypii* is considered the major pest of cucurbits. It is a polyphagous and destructive pest of more than twenty crop plant species. In hot regions, during the prolonged dry seasons, it produces large colonies on Cucurbitaceae, and it may survive on an ample variety of plant species, including cultivated and spontaneous Graminaceae [31]. In colder temperate regions, it is restricted to glasshouses, where it is a major pest. For some plant species, its direct feeding can cause serious damage to plant tissues, such as curled leaves and stunted shoots [32]. In zucchini plants, *A. gossypii* infestation causes a transcriptional up-regulation of genes underlying the biosynthesis of salicylic acid (SA) and of genes that modulate the SA-mediated defence response. As a consequence, aphids actively disperse on the plant, rather than starting their feeding activity where they were originally deposited, as observed in controls [33].

Although *A. gossypii* can cause direct damage to zucchini, the main damage is related to phytovirus transmission [34]. *A. gossypii* can transmit more than 50 phytoviruses, including non-persistent viruses of cucurbits, such as the Cucumber mosaic virus (CMV), the Zucchini yellow mosaic virus (ZYMV), the Papaya ringspot virus (PRSV), and the Watermelon mosaic virus (WMV) [31]. These viruses can infect zucchini plants and, as in the case of ZYMV, can cause 40 to 50% of yield losses [35]. Transmission of these viruses occurs during intracellular stylet punctures of aphids in epidermal or mesophyll cells, concomitant with saliva ejection [32].

In recent decades, the most used strategies to control aphid infestations and pathogens in zucchinis have been primarily focused on the selection of resistant genotypes [28,36,37] and on the use of pesticides [38,39]. Although pesticides may effectively reduce aphid populations in field, their use may improve the dispersion of viruses transmitted by aphids. This is due to the dispersive effect of some pesticides on aphids that survive the pesticide treatment [40,41]. Moreover, in literature, the development of resistance to insecticides in *A*. *gossypii* in several world regions is well reported [38,39,42,43,44].

The below-ground interactions between plants and microorganisms are very complex and it remains to be understood whether microbial biostimulants such as *Trichoderma* can be used to control harmful organisms. The aim of this study was to investigate the possibility of controlling zucchini pests and the most relevant virus diseases and powdery mildew in the field using the *Trichoderma harzianum* strain T22. The effects of inoculation of the commercial *T. harzianum* strain T22 on the arthropod community, on the above-mentioned plant diseases, and on the agronomic performance of zucchini squash was studied in detail for the first time in an experimental field.

## 2. Materials and Methods

### 2.1. Crop Cultivation 

The present study was performed in an experimental field located in Pignola (40°34′06.2″ N, 15°45′35.4″ E; 780 m above sea level), Potenza, Italy, during the period June to September 2022.

The soil was left fallow the year before the experiment and then ploughed to a depth of 25 cm, rotavated, and levelled before planting the crop. The soil characteristics are listed below: particles smaller than 2 mm in size, 935 g/kg; particles larger than 2 mm, 65 g/kg; apparent density, 1.294 kg/dm^3^; texture composition of sand, 481 g/kg; clay, 149 g/kg; silt, 370 g/kg at a depth of 0–30 cm. The content of total carbonate and total organic matter was of 16 g/kg and 32.8 g/kg, respectively. The composition of the soil was as follows: total N, 2 g/kg; P, 29 mg/kg; Ca 11.1 meq/100 g; Mg, 4.6 meq/100 g; Na, 1.8 meq/100 g; soil pH (H_2_O), 6.2. According to the world reference base for soil resources, the soil was a dystric cambisol (Bd68-2bc).

Zucchini seedlings (*Cucurbita pepo* L.) of the San Pasquale cultivar (Pagano Domenico & Figli, Scafati, Salerno, Italy) were used in this experiment. Zucchini plants placed in alveolate containers were purchased from a nursery and transplanted to the field on 6 June 2022. No fertilizers were used during the present experiment and the zucchini plants were not treated with any type of agrochemical during the entire field trial. Water irrigation was applied through the drip irrigation system.

### 2.2. Meteorological Data

The temperature and rainfall data recorded during the experiment are shown in Appendix A. During the period of interest, the average temperature was about 20 °C. The temperatures reached a maximum of 34 °C in August. The precipitations recorded in June, July, and in the first 15 days of August were very low. The Agrometeorological Service of the “Agenzia Lucana per lo Sviluppo e l’Innovazione in Agricoltura (ALSIA)” of the Basilicata Region provided the meteorological data for the area in which the experimental farm is located.

### 2.3. Experimental Design

The effects of *T. harzianum* T22 on diseases, insect community, and plant performance in zucchini plants were investigated. The first treatment consisted of non-inoculated zucchini plants (control), while the microbial biostimulant used in the present experiment to inoculate the plants was *Trichoderma harzianum* Rifai strain KRL-AG2 (T-22) (KOPPERT B.V., Berkel en Rodenrijs, the Netherlands), a purified strain which disperses in water.

The experiment was carried out on a strip of soil of about 40 m long and 14 m wide, divided into 6 plots of 22.5 m^2^ (9 m × 2.5 m) separated from each other by a strip 3 m wide left without plants. Thus, 3 plots treated with *T. harzianum* T22 and 3 control plots were obtained, alternating along the length of the field. The plants in each plot were manually transplanted on 6 June 2022 in 2 rows 1.8 m apart from each other. Each row was 8 m long with a plant spacing of 66 cm, with a total of 12 plants/row (24 plants/plot).

### 2.4. Fungal Inoculation

Before the experiments, the viability of the commercial formulation of *T. harzianum* T22 was evaluated in the laboratory by serial dilution. The dilutions were placed on Petri plates (9 cm in diameter) containing Potato Dextrose Agar (PDA) medium (Oxoid Ltd., Hants, UK) amended with the antibiotic streptomycin sulphate 40 mg/L (MerckKGaA, Darmstadt, Germany) until growth could be detected. As suggested by a previous study [45], the number of colony forming units (CFU) was counted after 24 h of incubation at 25 °C in the dark. 

Once the viability of the commercial product has been confirmed, zucchini seedlings were inoculated with *T. harzianum* T22 following the manufacturer’s instructions 5 days before transplantation. The alveolate containers with 24 seedlings were then watered with 3 g of commercial *T. harzianum* T22 (containing 1 × 10⁹ CFU/g of *T. harzianum* T22) dissolved in 3 litres of water. Each plant was watered with about 41 mL of the fungal suspension. The treatment was repeated after 4 days and then the seedlings were transplanted. A total of 72 inoculated zucchini plants and 72 non inoculated control plants were transplanted into the field. No fungal inoculation was performed after the transplantation.

In addition, 9 control and 9 treated plants were transplanted into pots and placed in a greenhouse to determine the presence of *T. harzianum* T22 in the roots. Twenty-four days after the last treatment, root samples were accurately collected and gently washed to remove soil residues. The colonization of *T. harzianum* T22 was confirmed by microscopy observations on squashed fine root hairs. In this study, the coloration of the zucchini hairy roots was performed following a rapid protocol set up using Tryptan blue 0.4% (Sigma Aldrich, Darmstadt, Germany) and commercial Pelikan blue (Hannover, Germany) dyes, glycerol 50%, and heating. To ensure the presence of *T. harzianum* T22 throughout the experiment, fungal colonization was also confirmed by strain isolations on PDA medium. Zucchini roots were sampled in the field 25 days (1 July), 54 days (30 July), and 85 days (30 August) after transplantation. For each sampling date, roots were collected from three control and three inoculated plants and transported to the laboratory. The roots were washed under running tap water to remove soil, then superficially sterilised using a 70% hydroalcoholic solution followed by a sodium hypochlorite solution at 1%. The roots were finally washed with sterile distilled water and dried on sterile paper. Each sample (a piece of root of about 2 cm^2^ in size) was placed on Petri plates with PDA medium amended with the streptomycin sulfate (0.05%). Plates were incubated at 25 ± 1 °C for 7 days and the presence of the fungus was then determined. 

### 2.5. Arthropod Sampling

During the first month after transplantation, the plants were very small, with few leaves and very few specimens of insects were found. From the second month after transplantation, the arthropod community on zucchini plants increased and was studied. An arthropod community survey was carried out by adopting two different sampling techniques: sampling of zucchini leaves, which mainly provided information on arthropod community colonising the plant, and capturing insects with colored pan trap sets, which mainly provided information on the winged arthropod community visiting the zucchini plant.

#### 2.5.1. Arthropod Sampling on Zucchini Leaves

To investigate the arthropod community on zucchini plants, within each plot, six plants were randomly sampled at 9:00 a.m., for a total of 36 plants/date. Sampled leaves, fully unfolded and of about 20 centimetres, were taken from the middle part of the zucchini plant. Each leaf was gently inserted in a transparent zip lock plastic bag (40 cm × 30 cm) and then cut at the insertion with it stem. This procedure allows an accurate sampling of small arthropods, also collecting the ones that drop and/or jump when the plant is touched, thus obtaining quantitative data on their abundance. The plastic bags were kept in darkness at 5 °C and transported to the laboratory for the identification of the arthropods. The collected arthropods were transferred to 50 mL Falcon tubes filled with ~30 mL of 70% hydroalcoholic solution and refrigerated at 4 °C until identification. The samples were then observed under a stereomicroscope. Arthropods from each sample were counted and classified at order, family, and, when possible, at the species level. Furthermore, the presence of damage caused by leaf miners on the leaves were noted and analysed. Five different leaf samples were carried out on 14 and 26 July, 9 and 26 August, and 8 September (that is, 38, 50, 64, 81, and 94 days after transplantation). 

#### 2.5.2. Arthropod Sampling with Coloured Pan Traps

Arthropods were also sampled using pan trap sets consisting of one blue, one yellow, and one white bowl. Pan trap is a passive sampling method that provides an ample return of data for relatively short periods of time and is particularly appropriate for faunal surveys [46], without a collector effect [47]. The traps were made by painting plastic bowls (17 cm in diameter, 4.5 cm deep), with blue (RAL standard colour codes: 5015) or yellow (RAL standard colour codes: 1023) acrylic paint sprays or left white. The pan traps were placed on the ground, in the middle of each experimental plot, as close as possible to the plants of each experimental plot. Each trap was filled with 400 mL of water and 4 mL of dishwashing detergent with no fragrance added to break surface tension. Traps were set out early at 8:00 a.m. and collected three days later, at the same time. In case of rain, pan traps were removed and the sampled specimens were not considered; the traps were replaced 24 h after rain stopped. Traps were collected in the order they were placed to ensure that all traps were available to insects for a similar time. Pan trap survey was carried out in four different data, with an interval of time of about two weeks among them: on 14 July, 1 and 16 August, and 8 September (that is, 38, 56, 71, and 94 days after transplantation). The arthropods were removed from the soap–water solution using a fine mesh colander and gently transferred with a soft paintbrush in 50 mL Falcon tubes, filled with 70% ethanol. Falcon tubes were stocked at 4 °C until the identification of the arthropods. The samples were then observed under a stereomicroscope. Subsequently, the arthropods of each sample were counted and classified according to their order, family, and, when possible, at the species level.

### 2.6. Evaluation of Diseases in Zucchini Plants 

The present study focused on the presence of zucchini viruses and powdery mildew since these are the most important zucchini diseases in the considered area. All plants were visually inspected for the presence/absence of diseases on 6 and 24 July, 4, 8, 19, and 26 August, and 7 September. The presence of virus (chlorosis, severe mosaic, deformation, blistering, and reduced leaf size) and/or powdery mildew (white powdery growth and subsequently spots or patches preferably on the leaf or on plant stems) symptoms was recorded. 

The zucchini viruses in plants were identified by enzyme-linked immunosorbent assay (ELISA), while the identification of the causal agent of the powdery mildew was done by microscopic observations.

#### 2.6.1. Evaluation of Zucchini Viruses in the Field

The presence and development of four of the most common zucchini plant viruses, Zucchini yellow mosaic virus (ZYMV), Papaya ringspot virus (PRSV), Cucumber mosaic virus (CMV) and Watermelon zucchini virus (WMV), were evaluated throughout the entire cultivation period. The symptoms of the viruses were visually observed in the field on all parts of the plant. To assess the degree of viral attack, each plant was examined individually and the degree of attack per plot was estimated using the following formula:$$Degree\;of\;attack\;\left(DA\%\right)=\frac{Number\;of\;symptomatic\;plants\;/\;plot\;}{Total\;number\;of\;plants/\;plot}\;\times \;100$$

The *DA*% for viruses was determined for all six plots during the experimental trial.

#### 2.6.2. ELISA Assay 

The presence of the four viruses investigated (CMV, ZYMV, PRSV, and WMV) was assessed twenty days before the last harvest. For each virus, the ELISA tests were performed using specific antibodies and kits (Loewe^®^ Biochemica GmbH, Sauerlach, Germany) following the manufacturer’s instructions. Briefly, a DAS ELISA [48] was performed using a polyclonal antiserum rabbit for each virus. Leaves and fruits from the experimental plot were placed in plastic bags, transported to the laboratory, and stored at 4 °C. For the ELISA test, the sap was extracted by homogenizing 1 g of sample in 10 mL of Conjugate/Sample buffer (ELISA kit) in plastic BIOREBA extraction bags (BIOREBA AG, Reinach, Switzerland) using a commercial homogenizer. Sap samples were collected in Eppendorf tubes and stored at −20 °C. The DAS ELISA assays consisted in coating the Nunc™ MicroWell 96-Well Microplates (ThermoFisher Scientific Inc., Waltham, MA, USA) plates (200 μL/well) with antigen-specific antibodies (IgG) 1:200 diluted in coating buffer, incubation of the plates at 37 °C for 4 h, followed by four manual washings with washing buffer at room temperature (RT), followed by samples application and overnight incubation at 4 °C. Subsequently, antibody–AP–conjugate application (200 μL/well) 1:200 diluted in conjugated buffer, incubation of the plates at 37 °C for 4 h, four washings, and enzymatic assays using substrate buffer added with 1 mg/mL of PNPP tablets were performed. The results were evaluated by comparing the visual reaction, determined as a yellow colour development, in the plate between the control (positive/negative) and samples. After 1 and 2 h of substrate incubation, plates were read photometrically at 405 nm wavelength using an ELISA Reader model A3 (DAS, Rome, Italy). All samples were run in duplicate.

#### 2.6.3. Powdery Mildew Evaluation Assay

The percentage of powdery mildew disease attack was assessed by field observations. All leaves and fruits from control and *T. harzianum* T22 treated zucchini plants were individually observed. To assess the percentage of powdery mildew attack in the field, the following scale was used: 0 (not infected) = 0% attack; 1 (low) = 1–25% infected tissue; 2 (medium) = 25.1–50% infected tissue; 3 (high) = 50.1–75% infected tissue; 4 (very high) = >75.1% infected tissue. Furthermore, to identify the possible pathogen causal agent responsible for the observed symptoms on zucchini plants, 25 symptomatic leaves and fruits were randomly collected from plants in the field and, on the same day, used to identify the causal agent of the powdery mildew in the laboratory. For species identification, conidia were directly obtained from the infected zucchini leaves and fruits collected in the field. The conidia were then observed under a light microscope (Axioscope, Zeiss, Germany) and also other morphological characteristics reported in literature for zucchini powdery mildew causal fungus were considered [49,50].

### 2.7. Evaluation of Plant Growth and Productivity

Plant growth was estimated by measuring the stem length of zucchini plants, excluding the leaf. This survey was carried out on four different data: on 22 June, 6 and 22 July, and 4 August. The stem length was measured in four plants per plot.

The zucchini fruits were first harvested on 9 July (33 days from transplanting) and were successively collected every two days until 22 August. From 22 August to 30 August, zucchini fruits were collected every four days. For each harvest, marketable fruits were counted and weighed for each experimental plot. The mean values of the weight of the zucchini fruits and the cumulative number and weight of the zucchini fruits per plant harvested from the plots inoculated with *T. harzianum* T22 and from the controls from 9 July to 30 August were then calculated.

### 2.8. Statistical Analysis

The number of arthropods sampled over time on the leaves, in the pan traps, and the data relating to the disease symptoms of the powdery mildew were analysed with a Poisson generalized linear mixed models (GLMMs) with a log-link function fitted with ML (maximum likelihood) and Laplace approximation. The discrete Poisson distribution best approximates the process that generated the observed data. The *p*-values for the differences between the treatments, sampling dates, and their interactions were obtained through analyses of deviance (Type II Wald chi-square tests). The following general model was applied:*Y = μ + Treatment + Date + Treatment × Date + Plot {Treatment {Date}} + ε*
where *Y* is the studied variable with a Poisson distribution, *Treatment* and *Date* are the fixed factors, and *Plot* is the random effect consisting of the three experimental plots nested in *Treatment* and *Date*. This model accounts for the non-independence of the data (pseudoreplication of measures) due to the different experimental plots (the random effect) that are part of the present design. 

Data on plant length, fruit weight, cumulative number, and cumulative weight of zucchini fruits per plant were analysed using linear mixed-effects models (LMMs) fitted with REML (restricted maximum likelihood). The homoscedasticity and normality assumptions for these ANOVAs were checked and met on these data. The *p*-values for the differences between the treatments, sampling dates, and their interactions were obtained through ANOVAs (type II Wald chi-square tests). To better appreciate the (possible) differences in fruit wight over time, in this analysis the sampling dates were grouped into 4 periods: 9–19 July, 20–31 July, 1–12 August and 13–25 August. The general model applied for these analyses was the same as applied for the analysis of the insect community.

The percentage of virus infected plants per plot was analysed using a linear model (LM) after an arcsine transformation of the data. The following model was applied:*Y = μ + Treatment + Date + Treatment × Date + ε*
where *Y* is the percentage (transformed) of virus-infected plants, and *Treatment* and *Date* (7 levels) are fixed effects. The *p*-values for were obtained by a factorial model ANOVA (Type II sum-square tests). 

To test for the influence of virosis on powdery mildew symptoms, the following model was applied:*Y = μ + Virus class + Treatment + Virus classes × Treatment + ε*
where *Y* is the quantification of the disease symptoms of the powdery mildew on a plant, *Virus class* (four levels of degree of virus attack, 0: 0 *DA*%, 1: 1–25 *DA*%, 2: 26–50 *DA*%, 3: 51–75 *DA*%, and 4: 76–100 *DA*%) and *Treatment* are fixed effects. The *p*-values were obtained by a factorial model ANOVA (Type II sum-square tests).

For all the analyses described so far, the model distributions were also chosen as the best fitting, based on AIC criteria [51] and the full models were presented. All statistical analyses were performed in R version 4.1.2 “Bird Hippie” [52], with lme4 [53], lmerTest [54] packages.

## 3. Results

### 3.1. Trichoderma harzianum T22 Inoculation

The zucchini plants transplanted into pots were inspected 24 days after inoculation to verify the success of the colonization of *T. harzianum* T22 in the roots. The 9 control and 9 inoculated plants were gently removed from the pots and photographed (Figure 1).

Compared to controls, the 9 zucchini plants inoculated with *T. harzianum* T22 showed an increased root development. In addition, light microscopy analyses showed that the colonization of the roots took place in all the 9 inoculated plants (100%). The presence of *T. harzianum* T22 was detected by the observation of the coloured intracellular structures of the fungus in the zucchini roots. The intracellular structures (the vesicles produced by the fungus) were not present in control samples but only in *T. harzianum* T22-treated samples (Figure 2).

In addition, *T. harzianum* T22 was isolated on PDA from all zucchini roots sampled in the treated plots 25 days, 54 days, and 85 days after transplantation. The presence of the fungus was not observed in the control Petri plates.

### 3.2. Arthropods Sampling

#### 3.2.1. Arthropod Sampling on Zucchini Leaves

Leaf samples were collected for observation of the arthropods in the laboratory. During this sampling period, 256 arthropod specimens were collected on zucchini leaves, of which 107 and 149 were obtained from plants with *T. harzianum* T22 and control, respectively. The arthropods on zucchini leaves belonged to the families Aphididae (one species identified: apterous morph of *Aphis gossypii*), Cicadellidae, Thripidae, Chrysomelidae, Gryllidae, Coccinellidae (adults), Syrphidae (identified as eggs or adults), Braconidae (adults), and Miridae. We also collected eggs of Lepidoptera, 9 individuals belonging to the order of Araneae, 5 individuals of *Tetranychus urticae*, and 5 leaf mines of *Liriomyza trifolii* Burges (Diptera, Agromyzidae). The abundances of Cicadellidae, Thripidae, Gryllidae, Chrysomelidae, Coccinellidae, Miridae, Syrphidae, Braconidae, Araneae, *T. urticae*, and leaf miners were very low during the whole sampling period (Appendix A) and consequently were excluded from the analysis.

The abundance of *A. gossypii* and of eggs of Lepidoptera is shown in Figure 3.

The GLMMs showed that the sampling dates influenced the abundance of apterous *Aphis gossypii* (χ^2^ = 20.1, df = 4, *p* < 0.001) and of eggs of Lepidoptera (χ^2^ = 10.2, df = 4, *p* < 0.05). The abundance of these insects was higher in July and then decreased in the following months. The abundance of *A. gossypii* was also affected by the treatment (χ^2^ = 6.2, df = 1, *p* < 0.05), with more apterous individuals collected on control plants. No significant differences between control and plants inoculated with *T. harzianum* T22 in the number of Lepidoptera eggs were observed (χ^2^ = 3.6, df = 1, *p* = 0.058). The interactions “treatment X date” were never found significant.

#### 3.2.2. Arthropod Sampling with Coloured Pan Traps

During the sampling period, 3925 arthropod specimens were collected with the pan traps, of which 2307 and 1618 were obtained from plants inoculated with *T. harzianum* T22 and control, respectively. The arthropods collected in the traps belonged to the families Aphididae (winged morphs of *A. gossypii*), Cicadellidae, Thripidae, Chrysomelidae, Gryllidae, Coccinellidae, Miridae, and to the order Lepidoptera, Hymenoptera (Ichneumonoidea and Chalcidoidea), and Araneae. The abundances of Lepidoptera, Coccinellidae, Staphylinidae, Gryllidae, and Miridae were very low, and they have not been considered for the analysis (Appendix A). 

The abundances of *A. gossypii*, Chrysomelidae, Thripidae, Cicadellidae, Hymenoptera parasitoids, and Araneae are shown in Figure 4.

The GLMMs showed that the abundances of all the arthropods collected with the pan trap sets in the experimental field were affected by the sampling dates (*p* < 0.001 in all cases). The abundance of arthropods was higher in July and then decreased in the following months. Significant differences between treatments were found for the abundance of winged *A. gossypii* (χ^2^ = 33.8, df = 1, *p* < 0.001), of Chrysomelidae (χ^2^ = 5.1, df = 1, *p* < 0.05), and of Hymenoptera parasitoids (χ^2^ = 61.9, df = 1, *p* < 0.001), with a higher number of insects collected in plots with zucchini inoculated with *T. harzianum* T22. The interaction “treatment X date” was only found significant for *A. gossypii* (χ^2^ = 439,322, df = 3, *p* < 0.001) and for the family of Cicadellidae (and χ^2^ = 10.1, df = 3, *p* < 0.05). Compared with the control, the abundance of aphids and Cicadellidae was higher on the 1 August on the *T. harzianum* T22 plots. 

### 3.3. Plant Diseases 

#### 3.3.1. Field Evaluation of Zucchini Viral Diseases 

The ANOVA performed on the data relating the viral infection gave significant differences among sampling dates (*F*_6,28_ = 127.6, *p* < 0.001) but not between treatment (*F*_1,28_ = 2.4, *p* = 0.13) or for the “treatment X date” interaction (*F*_6,28_ = 0.76, *p* = 0.61) as shown in Figure 5. The viral infections, in all plots, started on 24 July on both treated and untreated plants and continuously increased over time until the end of the cultivation period, reaching the 100% of infection at the beginning of September. On 7 September there was no difference in the viral symptoms observed in the field between the untreated (control) and treated (*T. harzianum* T22) plants per plot (Figure 5). 

Regarding the influence of the virosis on powdery mildew symptoms of the zucchini plants, it was observed that in plants with the same symptoms of virosis, the powdery mildew infection was more evident for the control plants compared with the *T*. *harzianum* T22 inoculated ones (Figure 6). Furthermore, the ANOVA performed on these data gave significant differences related to the virosis classes (*F*_4,514_ = 398.7, *p* < 0.001) and between treatments (*F*_1,514_ = 21.9, *p* < 0.001) but not for the “virosis classes X treatment” interaction (*F*_4,514_ = 0.6, *p* = 0.66), as shown in Figure 6. 

#### 3.3.2. ELISA Test for Viruses in Zucchini Plants

The results of the ELISA serological assay demonstrated that of the four most common zucchini viruses (CMV, ZYMV, PRSNV, and WMV), only one virus (CMV) was not present in the experimental field, while all others were detected. In particular, ZYMV and PRSNV were detected at 100%, WMV had a 45% of incidence in control plants and a 44% incidence in the *T. harzianum* T22-treated ones (Figure 7). In summary, our results showed that zucchini plants were infected by the three of four most common viruses and the viral incidence was not much different between the control and *T. harzianum* T22-inoculated plants.

#### 3.3.3. Powdery Mildew

Microscopic analysis showed that the causal agent of powdery mildew attack on zucchini plants in the field was closely similar to the *P. fusca*. These results are based on the morphological features reported in literature [49,50] for zucchini powdery mildew and the fibrosin bodies’ presence in the conidia.

The results regarding the symptoms of powdery mildew are shown in Figure 8. The symptoms were observed in the field after 19 August and the disease progressed in both controls and *T*. *harzianum* T22 plots, reaching 100% of infection on 7 September. Even if the disease symptom development was similar, a small delay was observed for the *T*. *harzianum* T22-treated plants compared to the control, at least in the initial and also during the disease development stages. However, the GLMMs showed that the symptoms of the powdery mildew were influenced by the sampling dates (χ2 = 200, df = 3, *p* < 0.001), but not by the treatment (χ2 = 0.79, df = 1, *p* = 0.37), nor by the interactions “treatment X date” (χ2 = 1.1, df = 3, *p* = 0.78) as shown in Figure 8.

### 3.4. Crop Sampling

#### 3.4.1. Plant Length

Figure 9 shows the mean values of the length of zucchini plants inoculated with *T. harzianum* T22 and control on the four sampling dates. 

For plant length, statistically significant differences were only found between sampling dates (χ^2^ = 204, df = 3, *p* < 0.001), indicating that plants inoculated with *T. harzianum* T22 or not inoculated have the same growth rate over time.

#### 3.4.2. Plant Productivity

Figure 10 shows the mean values of the weight of the zucchini fruit harvested from plants inoculated with *T. harzianum* T22 and from the control during the four sampling periods.

For fruit weight, statistically significant differences were only found among sampling periods (χ2 = 18.1, df = 3, *p* < 0.001), with heavier fruit produced during the first month. Even if the differences were not significant, during the first month plants inoculated with *T. harzianum* T22 showed a production of heavier fruit than controls.

Figure 11 shows the number and weight of fruits recorded after each harvest accumulatively from 9 July to 30 August from plants inoculated with *T. harzianum* T22 and from control ones.

The ANOVAs performed on these data show that the cumulative number of zucchini fruits/plant and the cumulative yield/plant were not affected by inoculation with *T. harzianum* T22 (χ^2^ = 2.55, df = 1, *p* = 0.11 and χ^2^ = 0.74, df = 1, *p* = 0.39, respectively). No significant interaction “date X treatment” was also found (χ^2^ = 4.6, df = 24, *p* = 0.99 and χ^2^ = 3.2, df = 24, *p* = 0.99, respectively). 

## 4. Discussion

The use of beneficial microbial species in agriculture as biocontrol agents and plant growth promoters has increased in recent decades [10]. Among them, fungi of the genus *Trichoderma* are the most widespread and effective [11,55,56]. *Trichoderma* fungi can antagonize plant pathogens through competition, antibiosis, and mycoparasitism mechanisms. *Trichoderma* is known to induce metabolic and physiologic changes in the colonized plants [57] by activating the plant SAR and/or by inducing systemic resistance against biotic and abiotic stress agents [11,20,21,22,23]. It is well accepted that plants have sophisticated defence strategies and when attacked by pathogens or pests activate signalling defence mechanisms modulated by jasmonic JA, SA, or ethylene phytohormones (ET) [11,21,58,59,60]. However, these pathways can crosstalk and their synergistic interactions can play a fundamental role in the ISR activation [61,62,63,64]. 

Recently, the possibility of using fungi of the genus *Trichoderma* as pest biocontrol agents has been emphasized [65]. Most studies on below ground–above ground interactions involving *Trichoderma* and plant pests have used the tomato as a model plant in the laboratory [12,14,16,17,18,19,66]. Few studies have verified, under far more complex field conditions, the role of *Trichoderma* fungi as a pest biocontrol agent [15,56,67].

In the context of horticultural crops, Cucurbitaceae is the second family in terms of economic relevance after Solanaceae [68], and zucchini is the most important economically and globally widespread species among the cultivated Cucurbitaceae [24]. The main phytosanitary problems of zucchini are aphids, phytoviruses, and powdery mildews [29,30]. The management of these pest and pathogen infections on zucchini crops has so far been based mainly on use of resistant varieties [28,36,37] and pesticides [38,39]. The effectiveness of *Trichoderma* spp. as biocontrol agents against zucchini fungal pathogens is confirmed by several studies, especially on *Fusarium* spp. [69,70]. Inoculations with *Trichoderma* spp. inhibited *F. oxysporum* infection stimulating plant metabolism and increasing the activities of stress-resistance enzymes [70].

On the contrary, laboratory and field studies that report the effectiveness of *T. harzianum* as a biocontrol agent against aphids and phytoviruses in zucchini are not available. In this study, we measured the natural evolution of pests and diseases in a zucchini field by comparing plots inoculated or not inoculated with *T. harzianum* T22. 

Throughout the cultivation period, *A. gossypii* was the only pest species worth mentioning. The highest occurrence on plants was observed on 26 July. Subsequently, the infestation decreased, probably due to the increase in temperature in mid-summer. Various studies on the ecology of aphid populations report a rapid population decline during the mid-summer, with host plants without aphids or with a lower abundance compared to the population abundance in early-summer and spring [71,72,73]. A major part of the aphids sampled on zucchini leaves were apterae, and in many cases, colonies were formed by just an adult aphid and a few nymphs. Interestingly, the abundance of aphids on leaves was significantly higher on control plants. *Trichoderma* is known to be involved in priming, the activation of plant defence prior to invasion, and up-regulated several Serine/threonine- and Leucine-rich repeat protein kinases that activate defence against pests [66]. *Trichoderma* colonization can generate a pre-alerted state of “priming” to face incoming pest attacks more efficiently [66,74], inhibiting the development and reproduction of aphids on the leaves of inoculated plants. In contrast, the winged aphids caught in pan traps were significantly more numerous in the plots inoculated with *T. harzianum* T22. These data seem to indicate that *T. harzianum* T22 makes zucchini plants more attractive to aphids, but this is followed by limited colony production. Winged aphids, while not producing colonies, could contribute to the spread of viruses.

Another significant result is the high number of Hymenoptera parasitoid captured in pan traps placed in plots inoculated with *T. harzianum* T22. The increased attractiveness to parasitoids and the reduced infestation of aphids, as a result of colonization by *Trichoderma*, confirm the results obtained in the laboratory, although with a different plant/aphid/parasitoid system [75]. *Trichoderma* influenced the quantity and quality of the volatile organic compound (VOC) blends released by plants [12]. The attractiveness to parasitoids is associated with an enhanced release of VOCs such as methyl-salicylate and β-caryophyllene, known to be among the most active compounds in promoting parasitoids flight orientation [12,75]. *Trichoderma* spp. promotes plant nitrogen uptake [20] giving the plant a higher nutritional value which can orient insects at the time of oviposition.

*Trichoderma harzianum* T22 failed to control zucchini pathogens investigated in the experimental field. Both viral diseases and powdery mildew equally attacked the control and *T. harzianum* T22-inoculated plants, starting from the end of July for viral diseases to middle of August in the case of powdery mildew. Furthermore, the severity of both the diseases worsened over time and the symptoms observed on zucchini plants changed from very mild to very strong, reaching the maximum peak at the beginning of September (expressed as 100% of infection). It may be useful to point out that laboratory experiments, testing the induction of resistance pathways by *Trichoderma* fungi, usually use young plants. In our experimental trial, in the field, it was observed that both viral infections and powdery mildew attack spread when plants had already begun to produce fruits. The ontogeny of resistance in plants has been approached with reference to insects [76,77,78], but still less is known about phytopathogens. For example, Vitti et al. [79], reported favourable effects of *T*. *harzianum* T22 in tomato seedling artificially inoculated with cucumber mosaic virus (CMV) in laboratory experiments. The authors showed that *T. harzianum* T22 was able to promote the induction of tomato defence responses against CMV and also demonstrated that this involves reactive oxygen species (ROS).

Another study by Shen et al. [28], investigating the dynamic distribution of *A. gossypii* on the incidence of viral disease in six zucchini cultivars, concluded that the ability of zucchini plants to resist aphids attack was not consistent with their capacity to resist viral diseases. Slow transformation rate varieties with a mild disease phenotype in the late growth stage showed strong resistance to the disease. 

We cannot exclude that the resistance induced by *Trichoderma* fungi observed in previous studies could be influenced by the phenological stage of the plant. The present study showed that there is no resistance effect in the field. Probably, the lack of resistance observed can be due to the higher costs of resistance for a plant that is already at the stage of fruit production. This aspect deserves future investigations. In fact, Shen et al. [28] showed that the disease resistance ability of zucchini plants always differed among the different growth stages.

Overall, no differences in terms of fruit yields were found between zucchini inoculated with *T. harzianum* T22 and control plants. This is in contrast to Hazef et al. [80], Elsisi [81], and El-Sharkawy et al. [82] who found an increase in the yield of zucchini plants under greenhouse and filed conditions due to the inoculation of *T. harzianum* T22. 

## 5. Conclusions

In this study, the effects of *Trichoderma harzianum* T22 as a biological control agent against zucchini the pests of were investigated for the first time under field conditions. The interaction among *T. harzianum* T22/zucchini plant/pests appeared to be complex. However, our results confirmed the ability of *T. harzianum* T22 to alter the arthropod community by increasing the attractiveness of zucchini to winged aphids and hymenopteran parasitoids. Unlike the outcomes of other studies conducted in the laboratory, a reduction in pathogen infestation was not observed in zucchini inoculated with *T. harzianum* T22. The discrepancies between our findings and the laboratory studies should be better investigated to understand how the abiotic factors affected the *Trichoderma*/plant interaction under open field conditions. It would also be interesting to investigate the ontogeny of the resistance mechanisms in function of the zucchini phenological stages. In our study, the presence of diseases, which were widely spread in the experimental field, and the lack of use of fertilizers and agrochemicals may have hidden the positive effect of *T. harzianum* T22 on plant production.

## Figures and Tables

**Figure 1 microorganisms-10-02242-f001:**
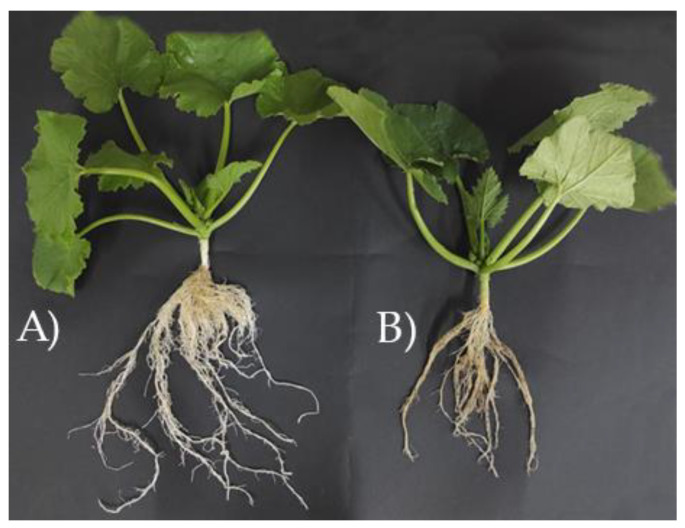
Zucchini plants 24 days after the inoculation. (**A**) *Trichoderma harzianum* T22 inoculated plant; (**B**) not inoculated control plant.

**Figure 2 microorganisms-10-02242-f002:**
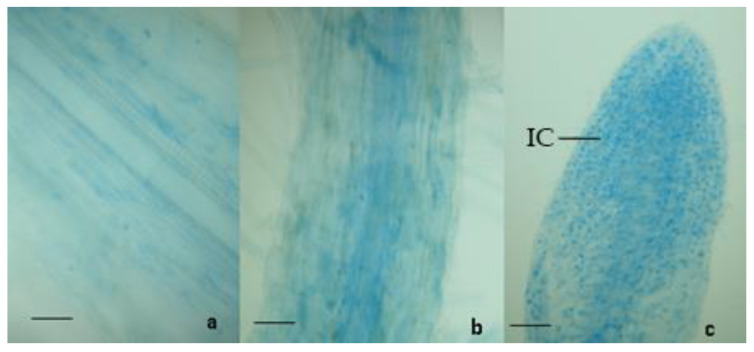
Light micrographs of zucchini roots stained with Tryptan blue and Pelikan blue dyes. (**a**,**b**): roots sections of control (not inoculated); (**c**): apex of primary root inoculated with *T. harzianum* T22, IC: intracellular structures; Bars = 100 μm.

**Figure 3 microorganisms-10-02242-f003:**
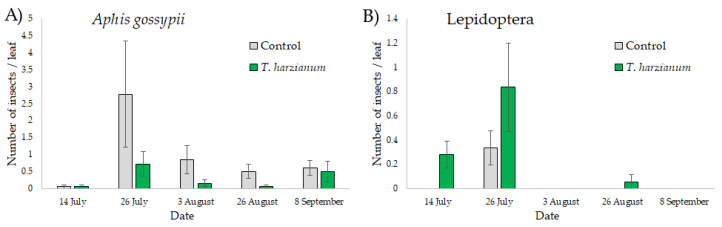
Mean values (±standard errors) of the number of apterous *Aphis gossypii* (**A**) and eggs of Lepidoptera (**B**) on zucchini leaves sampled from plants inoculated with *T. harzianum* T22 and control ones during the five sampling dates.

**Figure 4 microorganisms-10-02242-f004:**
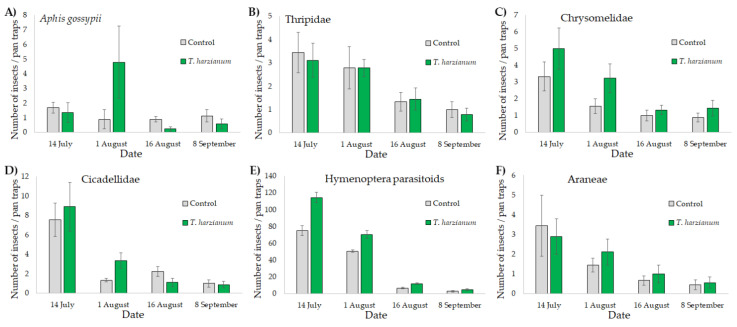
Mean values (± standard errors) of the number of winged *Aphis gossypii* (**A**), Thripidae (**B**), Chrysomelidae (**C**), Cicadellidae (**D**), Hymenoptera parasitoids (**E**), and Araneae (**F**) collected with pan trap sets placed near plants inoculated with *T. harzianum* T22 and control at the four sampling dates.

**Figure 5 microorganisms-10-02242-f005:**
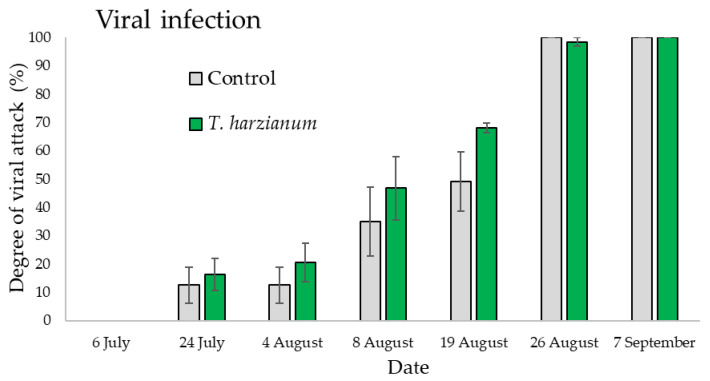
Mean degree of viral attack (±standard errors) in plots inoculated with *T. harzianum* T22 and control ones over time.

**Figure 6 microorganisms-10-02242-f006:**
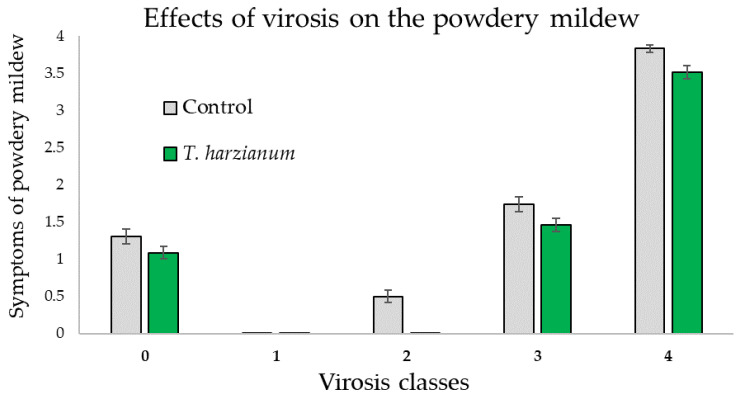
Mean values (±standard errors) of the symptoms the powdery mildew in relation to the virus classes in plants inoculated with *T. harzianum* T22 and control ones.

**Figure 7 microorganisms-10-02242-f007:**
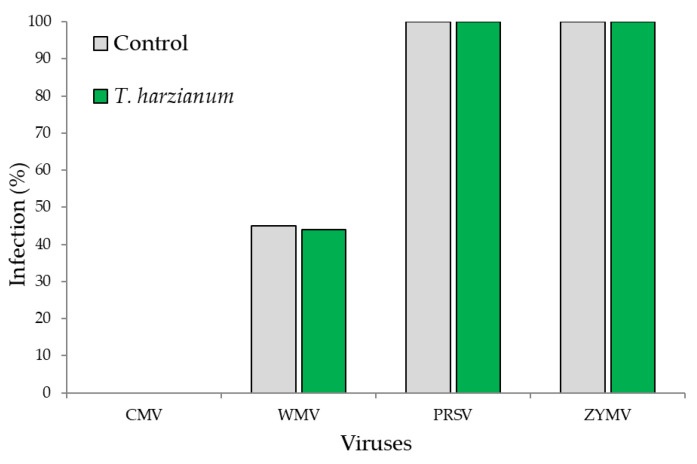
Percentage of infection for each virus determined by ELISA in zucchini plants.

**Figure 8 microorganisms-10-02242-f008:**
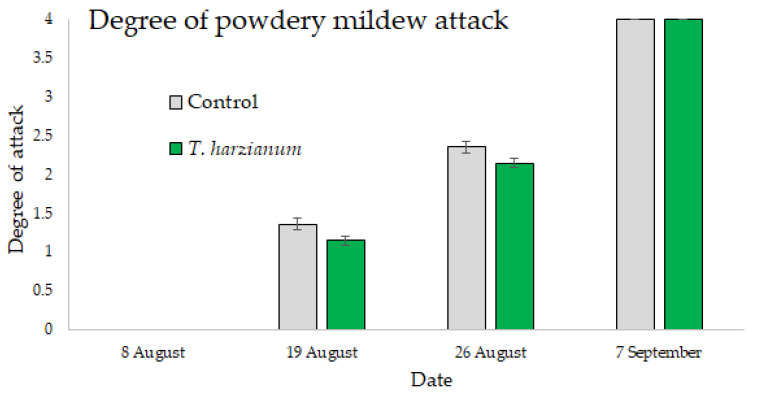
Mean values (±standard errors) of the powdery mildew degree of attack in plants inoculated with *T. harzianum* T22 and control over time.

**Figure 9 microorganisms-10-02242-f009:**
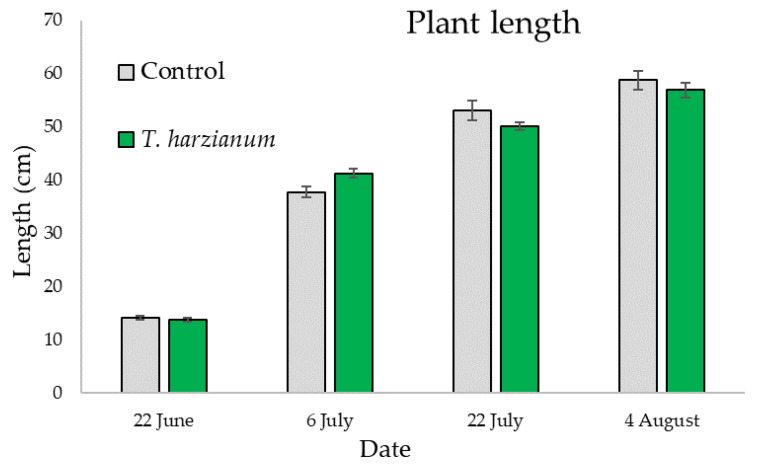
Mean values (±standard errors) of the length of zucchini plants inoculated with *T. harzianum* T22 and control on the four sampling dates.

**Figure 10 microorganisms-10-02242-f010:**
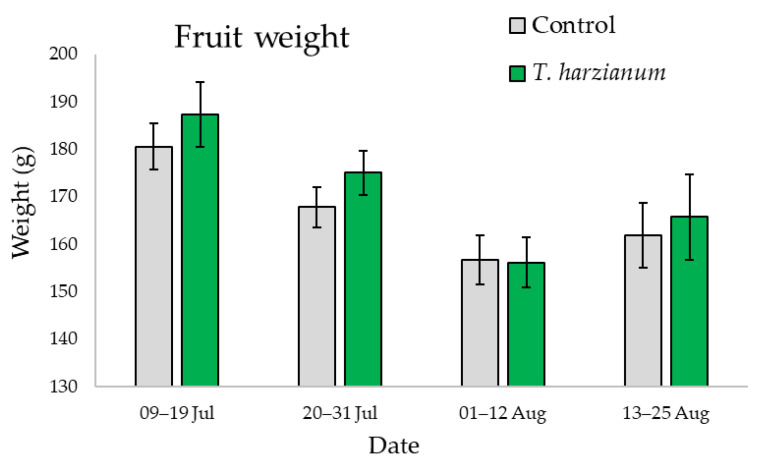
Mean values (±standard errors) of fruit weight from zucchini plants inoculated with *T. harzianum* T22 and control during the four sampling periods.

**Figure 11 microorganisms-10-02242-f011:**
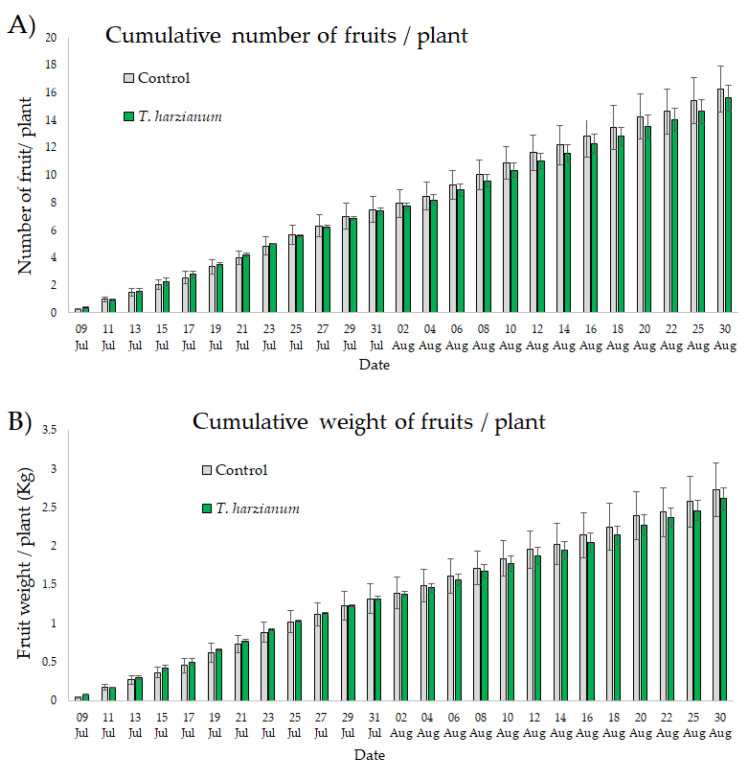
Mean values (±standard errors) of the cumulative number (**A**) and weight of zucchini fruits (**B**) recorded after each harvest from 9 July to 30 August from plants inoculated with *T. harzianum* T22 and from control.

## Data Availability

The data presented in this study are available on request from the corresponding author.

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
