# Peer review of "Effects of Below-Ground Microbial Biostimulant Trichoderma harzianum on Diseases, Insect Community, and Plant Performance in Cucurbita pepo L. under Open Field Conditions"

_microorganisms, 2022, doi:10.3390/microorganisms10112242_

Round 1

Reviewer 1 Report

The manuscript  "Effects of below-ground microbial biostimulant Trichoderma harzianum on diseases, insect community, and plant performance in Cucurbita pepo L. under open field conditions" contains information on the effect of  single-species bionoculans on plant defense. It contains information that can, in general, contribute to the understanding of induced defense in field conditions. However, the general speculations and the interpretation of the authors are usually too weak.

It is very important not to speculate on results when they are not statistically significant; for example, increased number of Lepidoptera eggs (lines 359-361).

I do not see the relationships between a higher number of parasitoids and a lower number of aphids, -for example in the month of July-, since the former were detected in aerial traps and the latter on the leaves. No symptoms of parasitism in the aphids present on the leaves were measured or detected.

Line 519 that refers to aphidophagous species is totally speculative, they are not detected or related to the decrease in aphids number/development. The same for lines 529-530, direct/indirect defenses were not measured either by some induced resistance gene or by the measurement of volatile compounds, for instance.

Author Response

We thank the Reviewer 1 for her/his very useful comments and efforts towards improving our manuscript. The following changes have been done as requested:

The manuscript  "Effects of below-ground microbial biostimulant Trichoderma harzianum on diseases, insect community, and plant performance in Cucurbita pepo L. under open field conditions" contains information on the effect of  single-species bionoculans on plant defense. It contains information that can, in general, contribute to the understanding of induced defense in field conditions.

R. Thank you very much, we appreciate your comments.

However, the general speculations and the interpretation of the authors are usually too weak.It is very important not to speculate on results when they are not statistically significant; for example, increased number of Lepidoptera eggs (lines 359-361).

R. As suggested by the reviewer, the correction was made in the text. The sentence has been replaced with the following:

“No significant differences between control and plants inoculated with T. harzianum T22 in the number of Lepidoptera eggs were observed (χ2 = 3.6, df = 1, P = 0.058).” (lines 386-388).

This sentence was also deleted from the discussion:

“As for chewing insects, a general increase in eggs of Lepidoptera was observed on plants inoculated with T. harzianum T22, although not significant. This finding is consistent with what was previously described for other field crops (Contreras-Cornejo et al., 2020; Caccavo et al., 2022).”

I do not see the relationships between a higher number of parasitoids and a lower number of aphids, -for example in the month of July-, since the former were detected in aerial traps and the latter on the leaves. No symptoms of parasitism in the aphids present on the leaves were measured or detected.

R. We agree with the referee, there is no relationship between the number of Hymenoptera parasitoids and the number of aphids sampled on the zucchini leaves, and no mummies were observed. This relationship is never mentioned in the manuscript. The parasitoids collected with pantraps belong to numerous families, and therefore they are not only parasitoids of aphids, but also parasitoids of eggs, pupae and adult insects. Unfortunately we have not been able to classify all the individuals sampled.

Line 519 that refers to aphidophagous species is totally speculative, they are not detected or related to the decrease in aphids number/development.

R. As suggested by the reviewer, this part has been removed: “… to the presence of aphidophagous species that colonized the zucchini plants …”

The same for lines 529-530, direct/indirect defenses were not measured either by some induced resistance gene or by the measurement of volatile compounds, for instance.

R. As suggested by the reviewer, this part has been removed “…due to the increased resistance, direct and/or indirect, induced in the plant.”

Date of this review

26 Oct 2022 10:07:28

R. We are sorry that we could not reply earlier, but these comments came after the first round of reviews (24 Oct).

Reviewer 2 Report

This is a very interesting work devoted to the possibility of using the well-known biopreparation based on Trichoderma harzianum T22 to protect Cucurbita pepo against dangerous pests for this plant species, such as aphids, other insect community, viroses and powdery mildews. The importance of this work is that the research was conducted in the field condition. However, most of the research to date has been carried out in laboratory conditions. Experiments were generally planned and performed in an appropriate manner. The results were generally presented in detail and in a clear form. Unfortunately, the conclusions are not optimistic, since it was formulated as follows 'the lack of use of fertilizers and agrochemicals may have hidden the positive effect of T. harzianum T22 on plant production'. The work should be published in Microorganisms after considering the comments below.

Line 35-36 it is 'Agrochemicals include pesticides, fungicides, ...' Pesticides are substances that are meant to control pests. Among others, pesticides include fungicides.

Line 51 a name error: it is Nezaria, it should be Nezara

Line 52 it should be jasmonic acid (JA) - the abbreviation used for the first time should be explained

Line 58 spp. - it should be not italic

Line 61 here is a mistake in the name of the fungus, it should be Golovinomyces cichoracearum

Line 61, line 434 Podosphaera xanthii - current name after Index Fungorum is Podosphaera fusca (Fr.) U. Braun & Shishkoff

Line 71 it should be salicylic acid (SA),

Line 105-109 why Kg, and not kg

Line 130 the used term Trichoderma harzianum strain Rifai KRL-AG2 (T-22) is illogical, although it is also used in other papers. Rifai is the author who described the new species Trichoderma harzianum Rifai (Rifai 1969), so the order should be Trichoderma harzianum Rifai strain KRL-AG2 (T-22). Of course, if the KOPPERT company uses a different term for its product, the name provided by the company must be used, but it should be checked.

Line 144 it is 40mg / L-1 it should be 40 mg / L

Line 148 it is cfu, in line 145 is given CFU

Line 156 should be reported if only the surface of root hairs was observed, whether sections were taken and whether internal endophytic colonization was examined

Line 172 am,  line 196 a.m. - the manner of recording should be standardized

Line 213 powdery fungal growth develops in cucurbits rather on both leaf surfaces

Line 225 needs correction, there are errors in the formula

Line 317 Chapter  '3.1. Trichoderma harzianum T22 Inoculation ' is poorly presented. It is not known in case of how many plants the root was longer in inoculated plants and, for example, by what average percentage. Figure 2 is not explained in the text what is the reader supposed to perceive in Fig 2a, b and what from 2c. Simple presentation of the photos doesn't explain much.

Line 324 it should be rather plant, not plants on Fig. 1

Line 333 T. - should be in italic

Figure 11A the word 'harzianum’ needs small correction

Line 434 (Castagne) U. Braun & Shishkoff should be crossed out, authors of names are given only the first time a given name is used in the text

Line 436 would be advisable to state whether only asexual morph or also sexual morph was observed

Line 509 here should be rather mildews, and not mildew

Line 512 it should be Fusarium spp.

Line 541 it is 'ovideposition' - shouldn't there be an oviposition?

Line 623 - it should be… Light Intensities

Line 704 it is Phitopathology, it should be Phytopathology

Line 741 it is Straint22, it should be Strain T22

Line 750 - there are mistakes in the names of the authors

Author Response

Reviewer 1

We thank the Reviewer 1 for her/his very useful comments and efforts towards improving our manuscript. The following changes have been done as requested by the reviewers:

Line 35-36 it is 'Agrochemicals include pesticides, fungicides, ...' Pesticides are substances that are meant to control pests. Among others, pesticides include fungicides.

As suggested by the reviewer, the correction was made in the text.

Line 51 a name error: it is Nezaria, it should be Nezara

As suggested by the reviewer, the correction was made in the text.

Line 52 it should be jasmonic acid (JA) - the abbreviation used for the first time should be explained

As suggested by the reviewer, the correction was made in the text.

Line 58 spp. - it should be not italic

As suggested by the reviewer, the correction was made in the text.

Line 61 here is a mistake in the name of the fungus, it should be Golovinomyces cichoracearum

As suggested by the reviewer, the correction was made in the text.

Line 61, line 434 Podosphaera xanthii - current name after Index Fungorum is Podosphaera fusca (Fr.) U. Braun & Shishkoff

As suggested by the reviewer, the corrections were made in the text.

Line 71 it should be salicylic acid (SA),

As suggested by the reviewer, the correction was made in the text.

Line 105-109 why Kg, and not kg

As suggested by the reviewer, the corrections were made in the text.

Line 130 the used term Trichoderma harzianum strain Rifai KRL-AG2 (T-22) is illogical, although it is also used in other papers. Rifai is the author who described the new species Trichoderma harzianum Rifai (Rifai 1969), so the order should be Trichoderma harzianum Rifai strain KRL-AG2 (T-22). Of course, if the KOPPERT company uses a different term for its product, the name provided by the company must be used, but it should be checked.

We agree with the reviewer, the correct name is Trichoderma harzianum Rifai strain KRL-AG2 (T-22) and consequentely the correction was made in the text. However, the commercial name is what we have previously reported in the manuscript, but it is not correct. We prefer to use the correct scientific name of the species, independently of the commercial label. Thank you again for your suggestion.

Line 144 it is 40mg / L-1 it should be 40 mg / L

As suggested by the reviewer, the correction was made in the text.

Line 148 it is cfu, in line 145 is given CFU

As suggested by the reviewer, the correction was made in the text.

Line 156 should be reported if only the surface of root hairs was observed, whether sections were taken and whether internal endophytic colonization was examined

The root hairs were squashed and then the internal colonization was microscopically observed. The correction was made in the text.

Line 172 am,  line 196 a.m. - the manner of recording should be standardized

As suggested by the reviewer, the correction was made in the text.

Line 213 powdery fungal growth develops in cucurbits rather on both leaf surfaces

As suggested by the reviewer, the correction was made in the text.

Line 225 needs correction, there are errors in the formula

As suggested by the reviewer, the correction was made in the text.

Line 317 Chapter  '3.1. Trichoderma harzianum T22 Inoculation ' is poorly presented. It is not known in case of how many plants the root was longer in inoculated plants and, for example, by what average percentage.

As suggested by the reviewer, we added these information: “… the colonization of the roots took place in all the 9 inoculated plants (100%)…”

Figure 2 is not explained in the text what is the reader supposed to perceive in Fig 2a, b and what from 2c. Simple presentation of the photos doesn't explain much.

As suggested by the reviewer, we added these information: “The presence of T. harzianum T22 was detected by the observation of the coloured structures of the fungus in the zucchini roots (Figure 2).”

Line 324 it should be rather plant, not plants on Fig. 1

As suggested by the reviewer, the correction was made in the text.

Line 333 T. - should be in italic

As suggested by the reviewer, the correction was made in the text.

Figure 11A the word 'harzianum’ needs small correction

As suggested by the reviewer, the correction was made in the figure.

Line 434 (Castagne) U. Braun & Shishkoff should be crossed out, authors of names are given only the first time a given name is used in the text

As suggested by the reviewer, the correction was made in the text.

Line 436 would be advisable to state whether only asexual morph or also sexual morph was observed.

Only the asexual morph was observed.

Line 509 here should be rather mildews, and not mildew

As suggested by the reviewer, the correction was made in the text.

Line 512 it should be Fusarium spp.

As suggested by the reviewer, the correction was made in the text.

Line 541 it is 'ovideposition' - shouldn't there be an oviposition?

As suggested by the reviewer, the correction was made in the text.

Line 623 - it should be… Light Intensities

As suggested by the reviewer, the corrections were made in the text.

Line 704 it is Phitopathology, it should be Phytopathology

As suggested by the reviewer, the corrections were made in the text.

Line 741 it is Straint22, it should be Strain T22

As suggested by the reviewer, the corrections were made in the text.

Line 750 - there are mistakes in the names of the authors

As suggested by the reviewer, the corrections were made in the text.

Reviewer 3 Report

The manuscript "Effects of below-ground microbial biostimulant Trichoderma harzianum on diseases, insect community, and plant performance in Cucurbita pepo L. under open field conditions" brings the results of productivity and incidence of pest and pathogens on zucchini after inoculation of T. harzianum. I believe the manuscript fails on the main hypothesis, which is show the beneficial effect of T. harzianum. Also, I missed fungal isolation to confirm the presence of T. harzianum throughout the experiment.

Specific comments:

I missed the definition of the acronyms in several points

l.59 - Homoptrera > Homoptera

l.130 - Was it used the purified strain or the diluted product?

l.147 - Inoculation methodology is not clear. Concentration tested was 1x10^9 CFU per g of viable spores? If the spores are viable, they should form colonies, so this unit does not make sense. After that, substrate was watered with a fungal suspension? Which volume per plant? This procedure occured before transplantion to the field? After that no more fungi was inoculated?

l.184 - Instead of dates, the authors could put number of days after inoculation or after transplantation

l.260 - Where did you obtain the conidia? Directly from the leaves and fruits, or there was a step of isolation?

l.329 - How can you confirm the blue spots are T. harzianum T22?

Figure 4 - Use bigger letters on the axes

l.413 - Virosis classes are refered to what?

l.462 - The decrease of plant productivity compared to control could be related to the absence of Trichoderma in the plant? The authors should have isolated the strain from the plant at different time intervals to ensure its presence throughout the experiment

Author Response

We thank the Reviewer 1 for her/his very useful comments and efforts towards improving our manuscript. The following changes have been done as requested:

The manuscript "Effects of below-ground microbial biostimulant Trichoderma harzianum on diseases, insect community, and plant performance in Cucurbita pepo L. under open field conditions" brings the results of productivity and incidence of pest and pathogens on zucchini after inoculation of T. harzianum. I believe the manuscript fails on the main hypothesis, which is show the beneficial effect of T. harzianum.

R. Our main hypothesis was not to show the beneficial effects of T. harzianum in field conditions, but to investigate its possible beneficial role:

“The aim of this study was to investigate the possibility of controlling zucchini pests and the most relevant virus diseases and powdery mildew in the field using the Trichoderma harzianum strain T22” (lines 92-94).

As stated by the reviever 2 (or reviewer 1 during the first round of revisions), “Unfortunately, the conclusions are not optimistic, since ……. the lack of use of fertilizers and agrochemicals may have hidden the positive effect of T. harzianum T22 on plant production”.

Also, I missed fungal isolation to confirm the presence of T. harzianum throughout the experiment.

R. The fungus was isolated both at the beginning, during, and at the end of the experiment. As suggested by the reviewer, we added this information:

“To ensure the presence of T. harzianum T22 throughout the experiment, fungal colonization was also confirmed by strain isolations on PDA medium. Zucchini roots were sampled in the field 25 days (1 July), 54 days (30 July), and 85 days (30 August) after transplantation. For each sampling date, roots were collected from three control and three inoculated plants and transported to the laboratory. The roots were washed under running tap water to remove soil, then superficially sterilised using a 70% hydroalcoholic solution followed by a sodium hypochlorite solution at 1%. The roots were finally washed with sterile distilled water and dried on sterile paper. Each sample (a piece of root of about 2 x 2 cm in size) was placed on Petri plates with PDA medium amended with the streptomycin sulfate (0.05%). Plates were incubated at 25 ± 1°C for 7 days and the presence of the fungus was then determined.” (lines 164-174).

“In addition, T. harzianum T22 was isolated on PDA from all zucchini roots sampled in the treated plots 25 days, 54 days, and 85 days after transplantation. The presence of the fungus was not observed in the control Petri plates.” (lines 356-358).

Specific comments:

I missed the definition of the acronyms in several points

R. As suggested by the reviewer, the corrections were made in the text (see also the answers to reviever 2 or reviewer 1 during the first round of revisions).

l.59 - Homoptrera > Homoptera

R. As suggested by the reviewer, the correction was made in the text.

l.130 - Was it used the purified strain or the diluted product?

R. The commercial product (purified strain) was used which disperses in water. This information has been added to the text: “…a purified strain which disperses in water” (line 131).

l.147 - Inoculation methodology is not clear. Concentration tested was 1x10^9 CFU per g of viable spores? If the spores are viable, they should form colonies, so this unit does not make sense. After that, substrate was watered with a fungal suspension? Which volume per plant? This procedure occured before transplantion to the field? After that no more fungi was inoculated?

R. As suggested by the reviewer, the text about inoculation methodology was modified. The concentration tested was 1x10^9 CFU per g of viable spores. However, before proceeding with the experiments, we tested the effective viability of the commercial formulation. This was done because it often happens that the commercial fungus spores are not viable (probably due to storage and transportation inefficiencies) or may be contaminated.

We added these information:

“Once the viability of the commercial product has been confirmed, zucchini seedlings were inoculated with T. harzianum T22 following the manufacturer's instructions 5 days before transplantation. The alveolate containers with 24 seedlings were then watered with 3 grams of commercial T. harzianum T22 (containing 1x10⁹ CFU/g of T. harzianum T22) dissolved in 3 litres of water. Each plant was watered with about 41 ml of the fungal suspension. The treatment was repeated after 4 days and then the seedlings were transplanted. A total of 72 inoculated zucchini plants and 72 non inoculated control plants were transplanted into the field. No fungal inoculation was performed after the transplantation.” (lines 148-156).

l.184 - Instead of dates, the authors could put number of days after inoculation or after transplantation

R. As suggested by the reviewer, we add this information:

“…that is, 38, 50, 64, 81, and 94 days after transplantation.” (line 199).

“..that is, 38, 56, 71, and 94 days after transplantation” (line 215).

However, we also left the dates as in a field experiment it is important to know when the observations were performed, since the environmental factors (temperature and rainfall) vary across the cultivation period.

l.260 - Where did you obtain the conidia? Directly from the leaves and fruits, or there was a step of isolation?

R. The condia were taken directly from the infected zucchini leaves and fruits from the field. We add this information:

“…were directly obtained from the infected zucchini leaves and fruits collected in the field.” (lines 275-276).

l.329 - How can you confirm the blue spots are T. harzianum T22?

R. The spots (that are vesicles produced by the fungus, as reported by other authors) were not present in control samples but only in T. harzianum T22 treated samples. We add this information: “…The intracellular structures (the vesicles produced by the fungus) were not present in control samples but only in T. harzianum T22 treated samples (Figure 2).” (lines 347-349).

Figure 4 - Use bigger letters on the axes

R. As suggested by the reviewer, the figure was modified

l.413 - Virosis classes are refered to what?

R. Virosis classes reffered to the degree of attack of the disease. See Materials and Methods, lines 325-326: “ … Virus class (four levels of degree of virus attack, 0: 0 DA%, 1: 1-25 DA%, 2: 26-50 DA%, 3: 51-75 DA%, and 4: 76-100 DA%).”

l.462 - The decrease of plant productivity compared to control could be related to the absence of Trichoderma in the plant? The authors should have isolated the strain from the plant at different time intervals to ensure its presence throughout the experiment

R. No significant differences in plant productivity between treatments were observed, even if the presence of Trichoderma was confirmed throughout the experiment (see the answer to the above comments).

Date of this review

26 Oct 2022 16:52:45

R. We are sorry that we could not reply earlier, but these comments came after the first round of reviews (24 Oct).

Reviewer 4 Report

The manuscript is very interesting and well designed, and the finding could help us to understand the roles of Trichoderma on plant performance under open field conditions. The manuscript is overall well written and clear to understand for the most part. However, I just have a few concerns that need to be addressed.

1. Please add the result in the Abstract.

2. Line 153-154, the presence of T. harzianum was determined with plants in the greenhouse, why not use the plants under the open field condition?

Author Response

Reviewer 2

We thank the Reviewer 2 for her/his very useful comments and efforts towards improving our manuscript. The following changes have been done as requested by the reviewers:

  1. Please add the result in the Abstract.

R: As suggested, the results were briefly reported in the Abstract: “Our results indicate that T. harzianum T22 makes zucchini plants more attractive to aphids and to Hymenoptera parasitoid but failed to control zucchini pathogens.”

  1. Line 153-154, the presence of T. harzianum was determined with plants in the greenhouse, why not use the plants under the open field condition?

R: The presence of T. harzianum was determined on plants placed in the greenhous inoculated at the same time and with the same experimetal procedure of the field ones. Under these conditions, the presence of the fungus should be indicative of the correct inoculum procedure, independently of the placement of the plants.

Reviewer 5 Report

In this study, authors had conducted field experiment that Zucchini were planted with or without the application of commercial fungal agent. The effect of fungal application on plant growth and fruit yield, the number and species of insect, virus infection were comprehensively investigated. These observations have their values for agronomic practice. However, the scientific logics and the mechanism, regarding to how the fungal inoculation regulated the developmental status and how the defensive ability against insects and pathogenic virus were built with the fungal inoculation, were unfortunately undisclosed. If the physiological or molecular investigation for the interaction and fungus and host plant could be further supplemented, it will be closer to the publication of this agronomic results in the journal named “Microorganisms”. Thus, the expression of insect-defense related genes in the host plant, the immunological study on the leaves of hostplant, or the pathogenic virus resistance related assay are suggested for the evidence or clue to disclose the scientific fact behind the phenomenon.

Author Response

Reviewer 3

In this study, authors had conducted field experiment that Zucchini were planted with or without the application of commercial fungal agent. The effect of fungal application on plant growth and fruit yield, the number and species of insect, virus infection were comprehensively investigated. These observations have their values for agronomic practice. However, the scientific logics and the mechanism, regarding to how the fungal inoculation regulated the developmental status and how the defensive ability against insects and pathogenic virus were built with the fungal inoculation, were unfortunately undisclosed. If the physiological or molecular investigation for the interaction and fungus and host plant could be further supplemented, it will be closer to the publication of this agronomic results in the journal named “Microorganisms”. Thus, the expression of insect-defense related genes in the host plant, the immunological study on the leaves of hostplant, or the pathogenic virus resistance related assay are suggested for the evidence or clue to disclose the scientific fact behind the phenomenon.

We thank the Reviewer 3 for her/his very useful comments, we will treasure them for the next studies. The molecular mechanisms induced by Trichoderma on plant defenses are well known, however, all the studies conducted so far refer to lab experiments. In lab experiments, it is easy to control all the environmental variables which can influence the outcomes of microbes-plants-insects-pathogens interactions; also, only one species of insect or of phytopathogens are considered separately. When instead field experiments are conducted, it is very difficult to separate all the effects influencing the plant responses to: environmental condition, multiple insect attacks, multiple phytopathogens attacks, as well as all their interactions. We agree that studies on molecular and physiological investigation on the interaction among fungus/plant/insects/phytopathogens are very interesting, but they should be conducted first in the lab considering many biotic and abiotic variables and, subsequently, one could think about their realization in such a complex context as an open field.

Round 2

Reviewer 1 Report

Although the study does not significantly increase global knowledge about the effect of bioinoculants on plant development, defense mechanisms, etc., the research design is appropriate and the description of the methods has been improved.

Reviewer 3 Report

Thank you for your answers. The authors added all the suggested modifications, making the comprehension more clear.

Reviewer 5 Report

Thank you for the reply and defence. I do agree with that. There is the difficulty to control the biotic and abiotic factor in the open field scale. However, I did not see substantial improvement regarding to my comments and suggestions on the logics behind the agronimic observation. I did not see also the additional evidence related to molecular or physiological data supporting the interaction between fungal innoculant and host plant.

Author Response

Thank you for the reply and defence. I do agree with that. There is the difficulty to control the biotic and abiotic factor in the open field scale. However, I did not see substantial improvement regarding to my comments and suggestions on the logics behind the agronimic observation. I did not see also the additional evidence related to molecular or physiological data supporting the interaction between fungal innoculant and host plant.

R. First of all, we would like to apologize for not previously modifying the manuscript in light of the comments provided by the reviewer on the logics behind our experiments. Unfortunately, we discovered the existence of the comments of the referee 5 (reviewer 3 during the first round of revisions) on 24 October, when uploading the new version of our manuscript, which had been accepted with minor revision.

As we have already explained, in field experiments, it is very difficult to separate all the effects influencing the plant responses to environmental conditions, multiple insect and phytopathogen attacks, as well as all their interactions. We are pleased that the referee agrees with us on these aspects.

Unfortunately, we do not have any molecular or physiological data on the interaction among fungus/plant/pests from the plants in our field experiments and it is impossible at this point to get these data. In order to obtain these molecular data, new experiments should be planned for the following zucchini season.  However, molecular or physiological data on the interaction between Trichoderma and host plant will be different, since it will be very improbable that the species, the levels of pest attacks, and the environmental variables will be the same as those recorded during the present experiment.

We agree that it is useful to discuss the molecular mechanisms behind our observations.

For example, some of us performed laboratory bioassays to assess the effect of Trichoderma on quantity and quality of volatile organic compounds. We are confident that the conclusions of this and other studies will be useful in discussing the results of the present experiment. We added these sentences in the Discussion section:

“It is well accepted that plants have sophisticated defence strategies and when attacked by pathogens or pests activate signaling defence mechanisms modulated by jasmonic JA, SA or ethylene phytohormones (ET) (Walling, 2000; Shoresh et al., 2010; Tucci et al., 2011; Ponzio et al., 2013; Macías-Rodríguez et al., 2020). However, these pathways can crosstalk and their synergistic interactions can play a fundamental role in the ISR activation (Perazzolli et al., 2008; Bari & Jones, 2009; Contreras-Cornejo et al., 2011; Salas-Marina et al., 2011).” (lines 528-532).

“Inoculations with Trichoderma spp. inhibited F. oxysporum infection stimulating plant metabolism and increasing the activities of stress-resistance enzymes (Li et al., 2019).” (lines 546-547).

“Trichoderma is known to be involved in priming, the activation of plant defense prior to invasion, and up-regulated several Serine/threonine- and Leucine-rich repeat protein kinases that activate defense against pests (Coppola et al., 2019). Trichoderma colonization can generate a pre-alerted state of “priming” to face incoming pest attacks more efficiently (Reimer-Michalski & Conrath, 2016; Coppola et al., 2019), inhibiting the development and reproduction of aphids on the leaves of inoculated plants.” (lines 561-566).

“Trichoderma influenced the quantity and quality of the volatile organic compound (VOC) blends released by plants (Battaglia et al., 2013). The attractiveness to parasitoids is associated with an enhanced release of VOCs such as methyl-salicylate and β-caryophyllene, known to be among the most active compounds in promoting parasitoids flight orientation (Battaglia et al., 2013; Coppola et al., 2017)” (lines 576-580).